# Decreased cerebrospinal fluid orexin levels not associated with clinical sleep disturbance in Parkinson's disease: A retrospective study

Takuya Ogawa[1,2], Yuta Kajiyama[1], Hideaki Ishido[3,4], Shigeru Chiba[3,5], Gajanan S. Revankar[1], Tomohito Nakano[6], Seira Taniguchi[1], Takashi Kanbayashi[3,5], Kensuke Ikenaka[1]*, Hideki Mochizuki[1]*

1 Department of Neurology, Osaka University Graduate School of Medicine, Osaka, Japan, 2 Department of Neurology, Morinomiya Hospital, Osaka, Japan, 3 International Institute for Integrative Sleep Medicine (WPI-IIIS), University of Tsukuba, Tsukuba, Japan, 4 Department of Neurology, Dokkyo Medical University Saitama Medical Center, Mibu, Japan, 5 Ibaraki Prefectural Medical Center of Psychiatry, Kasama, Japan, 6 Department of Neurology, Higashiosaka City Medical Center, Osaka, Japan

* ikenaka@neurol.med.osaka-u.ac.jp (KI); hmochizuki@neurol.med.osaka-u.ac.jp (HM)

**Data Availability Statement:** All data files are available from the Dryad database (DOI: 10.5061/dryad.905qfttq0).

## Abstract

Patients with Parkinson's disease (PD) often suffer from sleep disturbances, including excessive daytime sleepiness (EDS) and rapid eye movement sleep behavior disorder (RBD). These symptoms are also experienced by patients with narcolepsy, which is characterized by orexin neuronal loss. In PD, a decrease in orexin neurons is observed pathologically, but the association between sleep disturbance in PD and cerebrospinal fluid (CSF) orexin levels is still unclear. This study aimed to clarify the role of orexin as a biomarker in patients with PD. CSF samples were obtained from a previous cohort study conducted between 2015 and 2020. We cross-sectionally and longitudinally examined the association between CSF orexin levels, sleep, and clinical characteristics. We analyzed 78 CSF samples from 58 patients with PD and 21 samples from controls. CSF orexin levels in patients with PD (median = 272.0 [interquartile range = 221.7–334.5] pg/mL) were lower than those in controls (352.2 [296.2–399.5] pg/mL, p = 0.007). There were no significant differences in CSF orexin levels according to EDS, RBD, or the use of dopamine agonists. Moreover, no significant correlation was observed between CSF orexin levels and clinical characteristics by multiple linear regression analysis. Furthermore, the longitudinal changes in orexin levels were also not correlated with clinical characteristics. This study showed decreased CSF orexin levels in patients with PD, but these levels did not show any correlation with any clinical characteristics. Our results suggest the limited efficacy of CSF orexin levels as a biomarker for PD, and that sleep disturbances may also be affected by dysfunction of the nervous system other than orexin, or by dopaminergic treatments in PD. Understanding the reciprocal role of orexin among other neurotransmitters may provide a better treatment strategy for sleep disturbance in patients with PD.

**Funding:** YK and HM reports grants (Grant number 20FC1049) from the Research Committee of Central Nervous System Degenerative Diseases, Research on Policy Planning and Evaluation for Rare and Intractable Diseases, Health, Labour and Welfare Sciences Research Grants, the Ministry of Health, Labour and Welfare, Japan (https://www.mhlw.go.jp/english/); the funders had no role in study design, data collection and analysis, decision to publish, or preparation of the manuscript. TK reports grants (Grant number JP19dm0908001, JP20dm0107162, JP21zf0127005) from AMED (https://www.amed.go.jp/), and grants (19K08037) from JSPS KAKENHI Grant-in-Aid for Scientific Research (https://www.jsps.go.jp/j-grantsinaid/); the funders had no role in study design, data collection and analysis, decision to publish, or preparation of the manuscript.

**Competing interests:** The authors have declared that no competing interests exist.

## Introduction

Parkinson's disease (PD) and dementia with Lewy bodies (DLB) are common neurodegenerative diseases associated with the accumulation of Lewy bodies; both cause motor symptoms, including resting tremor, rigidity, bradykinesia, and postural instability. PD also causes sleep disturbances, including insomnia, sleep attacks, excessive daytime sleepiness (EDS), and rapid eye movement sleep behavior disorder (RBD), which often impair patients' cognition, psychiatric symptoms, and quality of life [1,2]. Sleep disturbances in patients with PD are not only due to the disease but also the treatment for PD, especially dopamine agonists (DAs) [3].

Orexin (also known as hypocretin) is a neuropeptide that regulates arousal and sleep. Deficiency of orexin neurons causes narcolepsy, characterized by EDS, sleep attacks, and cataplexy [4]. Narcolepsy-like symptoms are secondary to other neurological diseases, regardless of whether they occur owing to trauma, tumors, or immunological processes [4]. Since these sleep disturbances in PD are similar to those in narcolepsy, the association between orexin levels and PD has been studied [5]. Immunohistochemical analysis of the post-mortem brain tissues of patients with PD showed a decrease in orexin neurons, which further decreased with disease progression [6–8]. Moreover, the orexin levels in ventricular cerebrospinal fluid (CSF) also decrease in patients with advanced PD [7,9].

However, the implications of spinal CSF orexin levels in patients with PD remain inconclusive. In a study comparing orexin levels in patients with PD and controls, the orexin levels were lower in patients with PD than in controls [10], while others showed no significant difference [5,11]. A meta-analysis suggested that CSF orexin levels in patients with PD and DLB are lower than those in normal elderly individuals [12]. As the samples of those previous studies were small, studies with large samples are required to validate the difference in CSF orexin levels between PD and other conditions [13].

Regarding the clinical characteristics of PD, a previous study suggested that CSF orexin levels decrease with longer disease duration [14], while others showed no correlation [15]. EDS is one of the most common sleep disturbances in both PD and narcolepsy; however, studies focusing on EDS using the Epworth Sleepiness Scale (ESS) showed no significant correlation [16,17]. RBD is also seen in PD and narcolepsy, but the relationship between RBD and CSF orexin in PD is controversial. A study reported no significant differences between patients with PD with and without RBD [18], while the relationship between CSF orexin levels and the severity of RBD in patients with DLB has been demonstrated [19]. Furthermore, the effect of PD treatment using DAs on CSF orexin levels has been shown in vitro [20]; however, only one small study has suggested the effect in vivo [14].

Regarding the longitudinal disease progression, one study revealed that orexin levels decrease over years [21]. To our knowledge, no study has compared the changes in CSF orexin levels with disease progression in patients with PD. Thus, it is unknown whether CSF orexin levels can affect future clinical characteristics of PD.

Since previous studies had insufficient samples or used inconsistent methods of clinical assessment [10,11,13,14,16,18,19,21], the diagnostic or prognostic value of the CSF orexin level in PD lacks evidence. In this study, we measured orexin levels in the CSF samples obtained from patients with PD to clarify the role of orexin as a biomarker of PD, in the following particulars: (i) its difference between PD and other diseases, (ii) its association with the severity of sleep disturbances and other clinical characteristics, and (iii) its relationship with and predictive value of the longitudinal course of PD. This study may provide precise insights into sleep disturbance and an appropriate treatment target in PD.

## Materials and methods

### Participants

We selected two sets of participants from our previous cohort study [22,23]: (i) participants for longitudinal data and (ii) participants for extraction of cross-sectional data for analysis of the association between CSF orexin levels, sleep characteristics, and DA use in the treatment of PD (Fig 1). To obtain longitudinal data, all participants registered in our cohort study between April 2017 and August 2019 were recruited. The inclusion criteria were: (i) a confirmed diagnosis of PD according to the Movement Disorder Society Diagnostic Criteria for PD and (ii) evaluation of CSF at two or more different time periods. To obtain cross-sectional data, participants registered in our cohort study between April 2015 and April 2020 were recruited. The inclusion criteria were: (i) both confirmatory and probable diagnoses of PD, (ii) disease duration within eight years of motor onset, and (iii) medication with a single DA use or without DAs. Participants taking orexin receptor antagonists were excluded from both data sets. All those without adequate clinical evaluation results with the ESS [24] and Rapid Eye Movement Sleep Behavior Disorder Questionnaire (RBDQ) [25] were also excluded from the comparison of clinical characteristics among PD patients. Age- and sex-matched controls were selected from patients without neurodegenerative diseases.

Patients with PD were categorized according to three criteria: (i) the EDS- group with ESS scores <11 and the EDS+ group with ESS scores ≥11 according to the original report [24]; (ii) the RBD- group with RBDQ scores <5 and RBD+ group with RBDQ scores ≥5 according to the original report [25]; and (iii) the RTG group with patients on rotigotine, ROP group with patients on ropinirole, PPX group with patients on pramipexole, and no DA group with patients who were never on DAs.

Longitudinal changes were calculated from the clinical and CSF data of patients with PD obtained during different periods. In patients with PD examined more than three times, longitudinal changes were calculated from each dataset using the initial data.

### Ethics

This study was conducted as part of a prospective and exploratory study for disease-specific biomarkers and objective indicators in neurodegenerative diseases (UMIN-CTR: UMIN000036570) and the Osaka University Longitudinal Biomarker Study for Neuromuscular Diseases. This study was conducted in accordance with the Declaration of Helsinki and Ethical Guidelines for Medical and Health Research Involving Human Subjects endorsed by the Japanese government. The Ethics Committee of Osaka University Graduate School of Medicine approved this study (approval numbers: 13471 and 19089). All patients, including the controls, were informed about this study and provided written consent.

### Clinical assessment

To evaluate the association between CSF orexin levels and sleep disturbances, we obtained clinical characteristics, including age, sex, disease duration, and levodopa equivalent daily dose of anti-Parkinson drugs. Sleep disturbance was assessed using the ESS and RBDQ. To assess the general clinical features of PD, we obtained the Hoehn and Yahr scale (HYscale) [26], Movement Disorder Society—Unified Parkinson's Disease Rating Scale (MDS-UPDRS) [27], and Parkinson's Disease Questionnaire (PDQ-39) results [28]. Cognitive dysfunction was assessed using the Mini-Mental State Examination (MMSE) [29] and frontal assessment battery (FAB) [30]. To assess the non-motor symptoms of PD, we obtained the olfactory

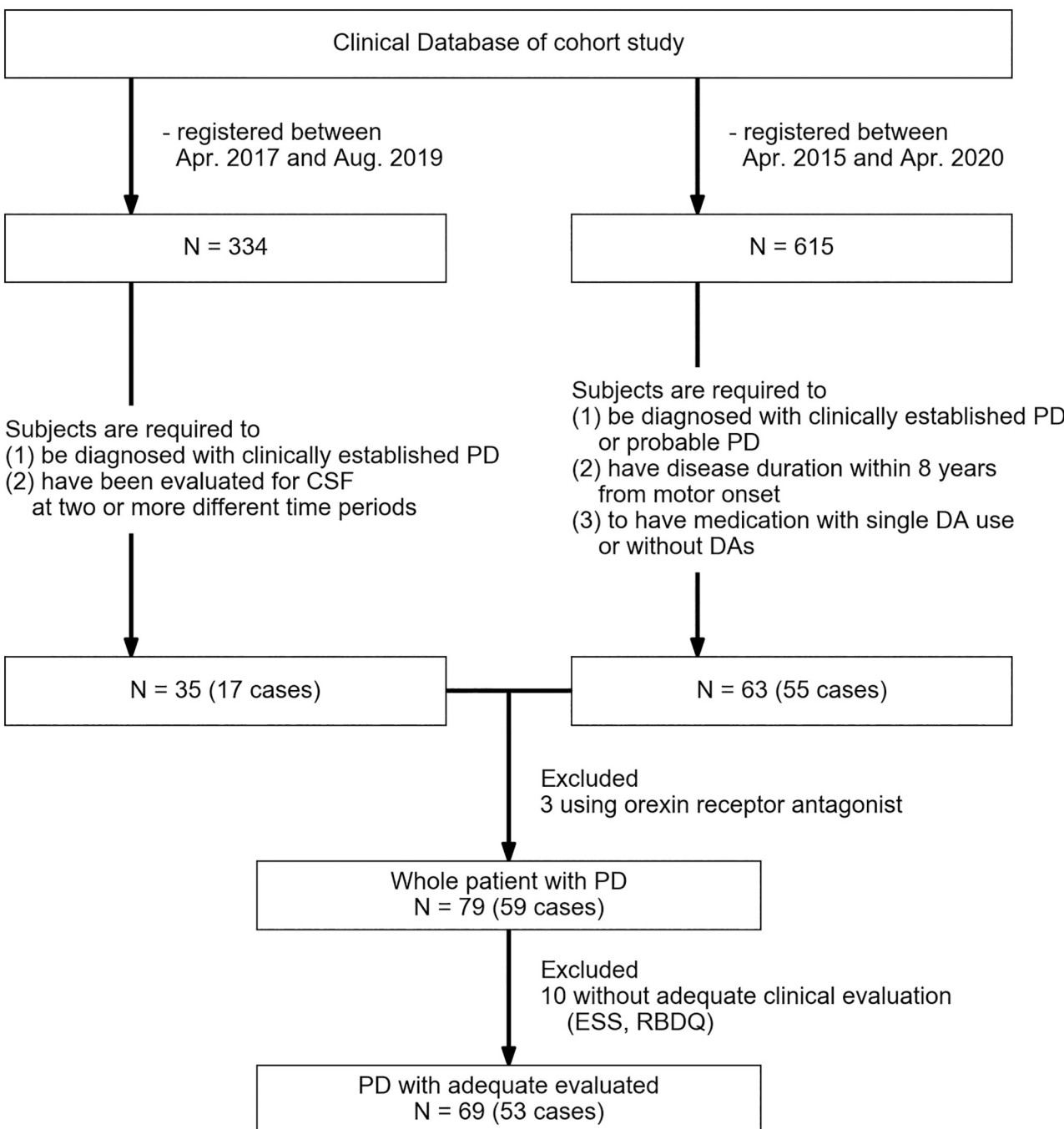

**Fig 1. Flowchart illustrating the study design.** Confirmatory or probable PD were diagnosed according to the Movement Disorder Society Diagnostic Criteria for PD. N, number of samples; PD, Parkinson's disease; RBDQ, Rapid Eye Movement Sleep Behavior Disorder Questionnaire; DA, dopamine agonist.

identification score, Scales for Outcomes in Parkinson's Disease-Autonomic (SCOPA-AUT) [31], Geriatric Depression Scale (GDS-15) [32], and Apathy Scale (AS) [33]. The olfactory identification score was measured using a card-type kit ("Open Essence"; FUJIFILM Wako Pure Chemical Corporation, Osaka, Japan).

### Orexin measurements

CSF was collected by lumbar puncture in the morning after overnight fasting, keeping the usual medications. The CSF samples were centrifuged at $400 \times g$ for 10 min at 4˚C, and aliquots were stored at −80˚C before analysis. CSF orexin levels were measured using a radioimmunoassay kit (Phoenix Pharmaceuticals, Belmont, CA, USA) by the International Institute for Integrative Sleep Medicine at the University of Tsukuba.

### Statistical analysis

Python version 3.7.6 was used to perform all statistical analyses. The orexin levels and clinical characteristics between the groups were compared using the Mann–Whitney U test, except for sex, using the chi-square test. Correlations between orexin and age, disease duration, HYscale, MDS-UPDRS, PDQ-39, MMSE, FAB, ESS, RBDQ, olfactory identification score, SCOPA-AUT, GDS-15, and AS were evaluated using Spearman's rank correlation coefficient. Longitudinal changes in orexin levels were evaluated using the Wilcoxon signed-rank test. The effect size was calculated by dividing Z statistics by the square root of the sample size. Statistical significance was set at 0.05, with Bonferroni correction, if necessary.

Based on previous studies, we considered age, sex, disease duration, sleep disturbance, severity of PD, and cognitive function as confounding factors. Therefore, multivariable analysis was conducted using a linear regression model with CSF orexin levels as a dependent variable, and clinical characteristics including age, sex, disease duration, ESS, MDS-UPDRS part 3, and MMSE as independent variables. Before analysis, all clinical characteristics were standardized. Participants with missing data for any of these clinical characteristics were excluded before correlation analysis and multiple linear regression analysis.

## Results

### Clinical characteristics of participants

The clinical characteristics of the study participants are shown in Table 1. A total of 78 CSF samples from 58 patients with PD were analyzed in this study, including 16 patients with two longitudinal samples and two patients with three samples taken at different time periods. The control group consisted of 16 patients with idiopathic normal pressure hydrocephalus, two patients with cervical spondylotic myelopathy, one patient with psychosomatic disease, one patient with complex regional pain syndrome, and one patient without neurological or psychological disease. The clinical characteristics of each group are presented in Table 1.

### Orexin levels of patients with Parkinson's disease

First, we examined CSF orexin levels in patients with PD and controls (Fig 2 and Table 1). The median CSF orexin level of patients with PD (272.0 [interquartile range (IQR) = 221.7–334.5] pg/mL) was significantly lower than that of controls (352.2 [296.2–399.5] pg/mL) (Z statistic = 2.72, p = 0.007, effect size r = 0.270).

### Orexin levels and clinical characteristics

We analyzed 68 CSF samples of patients with PD with adequate data by group comparisons. When groups were compared, no significant differences were noted in CSF orexin levels between the EDS+ (278.7, IQR = [235.3–329.2] pg/mL) and EDS- (260.4 [203.6–309.7] pg/mL) groups, and the RBD+ (263.0 [227.8–306.3] pg/mL) and RBD- (271.1 [217.0–336.3] pg/mL) groups. Moreover, the RTG (275.2 [236.3–338.5] pg/mL), ROP (271.8 [257.9–322.2] pg/mL),

**Table 1. Characteristics of participants\*.**

| | N | PD | N | Control | N | EDS+ | N | EDS- | N | RBD+ | N | RBD- | N | RTG | N | ROP | N | PPX | N | no DA |
|---|---|---|---|---|---|---|---|---|---|---|---|---|---|---|---|---|---|---|---|---|
| N(p) | | 58 | | 20 | | 21 | | 32 | | 20 | | 33 | | 9 | | 6 | | 8 | | 34 |
| N(s) | | 78 | | 21 | | 25 | | 43 | | 24 | | 44 | | 11 | | 7 | | 8 | | 40 |
| Sex** | | 42: 36 | | 12: 9 | | 14: 11 | | 23: 20 | | 15: 9 | | 22: 22 | | 4: 7 | | 4: 3 | | 3: 5 | | 24: 16 |
| Age | 78 | 69.0 [62.2–74.0] | 21 | 72.0 [68.0–75.0] | 25 | 67.0 [64.0–73.0] | 43 | 69.0 [61.0–74.0] | 24 | 67.0 [64.8–74.0] | 44 | 69.0 [61.0–72.2] | 11 | 71.0 [63.0–73.5] | 7 | 63.0 [57.0–69.5] | 8 | 67.5 [64.0–71.0] | 40 | 69.0 [61.0–74.0] |
| Duration | 78 | 5.0 [3.0–7.0] | | NA | 25 | 5.0 [4.0–7.0] | 43 | 4.0 [2.0–6.0] | 24 | 6.0 [4.8–8.0] | 44 | 4.0 [2.0–6.0] *c | 11 | 6.0 [4.5–7.5] | 7 | 5.0 [4.0–7.0] | 8 | 5.5 [4.8–6.2] | 40 | 4.0 [2.0–5.2] |
| ESS | 68 | 8.0 [5.0–13.0] | | NA | 25 | 16.0 [13.0–18.0] *b | 43 | 5.0 [3.0–8.0] | 24 | 13.5 [8.8–18.5] | 44 | 6.5 [3.8–10.0] *c | 11 | 13.0 [9.0–18.5] *d | 7 | 10.0 [9.5–12.0] | 8 | 8.0 [6.5–11.0] | 40 | 6.0 [3.0–10.0] |
| RBDQ | 69 | 3.0 [1.0–5.0] | | NA | 25 | 5.0 [3.0–8.0] *b | 43 | 2.0 [1.0–4.0] | 24 | 6.0 [5.0–8.2] | 44 | 1.0 [1.0–3.0] *c | 11 | 5.0 [4.5–6.5] | 7 | 1.0 [0.5–2.5] | 8 | 4.0 [1.0–5.0] | 40 | 2.0 [1.0–4.2] |
| HYscale | 77 | 2.0 [2.0–3.0] | | NA | 25 | 2.0 [2.0–3.0] | 43 | 2.0 [2.0–3.0] | 24 | 2.0 [2.0–3.0] | 44 | 2.0 [2.0–3.0] | 11 | 3.0 [2.0–3.0] | 7 | 3.0 [2.0–4.0] | 8 | 2.0 [2.0–2.2] | 40 | 2.0 [2.0–3.0] |
| UPDRS (I) | 74 | 9.5 [5.0–14.0] | | NA | 25 | 12.0 [9.0–18.0] *b | 42 | 8.0 [3.0–12.8] | 24 | 14.0 [11.0–19.0] | 43 | 8.0 [3.0–11.0] | 11 | 18.0 [13.0–20.5] *d | 7 | 11.0 [8.0–13.0] | 8 | 8.5 [5.8–9.5] | 39 | 8.0 [3.5–12.0] |
| UPDRS (II) | 74 | 12.0 [7.0–18.0] | | NA | 25 | 16.0 [13.0–18.0] *b | 42 | 9.0 [6.0–17.5] | 24 | 18.0 [15.0–23.5] | 43 | 8.0 [5.0–15.5] | 11 | 18.0 [15.5–24.0] *d | 7 | 15.0 [10.5–20.0] | 8 | 11.0 [5.2–12.0] | 39 | 9.0 [5.5–17.0] |
| UPDRS (III) | 76 | 26.5 [19.0–37.5] | | NA | 25 | 28.0 [22.0–34.0] | 43 | 26.0 [16.0–40.0] | 24 | 33.0 [22.0–41.5] | 44 | 23.0 [13.0–32.8] | 11 | 31.0 [20.0–37.0] | 7 | 27.0 [24.0–41.0] | 8 | 20.0 [12.5–21.5] | 40 | 28.0 [18.8–37.5] |
| UPDRS (IV) | 74 | 0.0 [0.0–5.8] | | NA | 25 | 2.0 [0.0–7.0] *b | 42 | 0.0 [0.0–4.8] | 24 | 5.5 [0.0–8.0] | 43 | 0.0 [0.0–2.5] | 11 | 8.0 [2.5–8.5] *d | 7 | 2.0 [1.5–6.0] | 8 | 0.0 [0.0–1.2] | 39 | 0.0 [0.0–3.5] |
| PDQ-39 | 68 | 32.5 [22.8–61.0] | | NA | 25 | 61.0 [45.0–76.0] *b | 42 | 25.5 [15.5–33.8] | 24 | 61.0 [47.2–77.5] | 43 | 27.0 [16.0–33.5] | 11 | 59.0 [51.0–75.0] *d | 7 | 30.0 [29.0–58.0] | 8 | 37.0 [27.8–47.8] | 39 | 27.0 [16.0–59.5] |
| MMSE | 77 | 27.0 [25.0–30.0] | | NA | 25 | 27.0 [22.0–29.0] | 43 | 28.0 [26.0–30.0] | 24 | 26.0 [22.8–29.0] | 44 | 29.0 [26.0–30.0] | 11 | 26.0 [23.0–29.0] | 7 | 30.0 [27.5–30.0] | 8 | 27.0 [26.0–30.0] | 40 | 28.0 [26.0–30.0] |
| FAB | 76 | 15.0 [13.0–17.0] | | NA | 25 | 15.0 [12.0–17.0] | 43 | 15.0 [13.0–17.0] | 24 | 14.5 [11.8–16.0] | 44 | 15.0 [13.0–18.0] | 11 | 14.0 [12.0–14.5] | 7 | 16.0 [14.0–17.5] | 8 | 15.0 [13.8–16.2] | 40 | 15.0 [13.8–18.0] |
| Olfactory | 74 | 3.5 [2.0–5.0] | | NA | 24 | 3.0 [2.0–5.0] | 42 | 4.0 [3.0–5.0] | 23 | 3.0 [1.5–4.0] | 43 | 4.0 [3.0–5.5] | 10 | 2.5 [2.0–4.8] | 7 | 3.0 [3.0–3.5] | 8 | 4.5 [3.0–6.2] | 39 | 4.0 [3.0–5.0] |
| SCOPA-AUT | 70 | 14.5 [8.0–22.0] | | NA | 25 | 18.0 [13.0–27.0] *b | 43 | 12.0 [6.0–19.0] | 24 | 22.0 [16.8–28.2] | 44 | 12.0 [6.0–16.2] | 11 | 14.0 [9.5–24.5] | 7 | 16.0 [13.5–21.5] | 8 | 21.5 [15.2–24.2] | 40 | 12.5 [6.0–18.0] |
| GDS-15 | 69 | 4.0 [2.0–7.0] | | NA | 25 | 6.0 [4.0–10.0] *b | 43 | 3.0 [2.0–5.0] | 24 | 6.0 [3.8–8.2] | 44 | 3.0 [2.0–5.2] | 11 | 7.0 [2.0–9.0] | 7 | 4.0 [2.0–7.0] | 8 | 4.0 [2.2–5.0] | 40 | 4.0 [2.0–6.0] |
| AS | 69 | 14.0 [8.0–18.0] | | NA | 25 | 15.0 [9.0–23.0] | 43 | 14.0 [7.0–17.0] | 24 | 14.0 [8.8–22.0] | 44 | 14.0 [7.0–17.0] | 11 | 13.0 [8.5–17.5] | 7 | 14.0 [9.5–24.0] | 8 | 14.0 [10.0–15.0] | 40 | 15.0 [7.0–18.2] |
| Orexin | 78 | 272.0 [221.7–334.5] *a | 21 | 352.2 [296.2–399.5] | 25 | 278.7 [235.3–329.2] | 43 | 260.4 [203.6–309.7] | 24 | 263.0 [227.8–306.3] | 44 | 271.1 [217.0–336.3] | 11 | 275.2 [236.3–338.5] | 7 | 271.8 [257.9–322.2] | 8 | 231.7 [207.7–278.8] | 40 | 269.8 [204.6–318.6] |

PD, Parkinson's disease; EDS, excessive daytime sleepiness; RBD, rem eye movement sleep behavior disorder; RTG, rotigotine; ROP, ropinirole; PPX, pramipexole; no DA, no use of dopamine agonist; N(p), number of patients; N(s), number of samples; HYscale, Hoehn and Yahr scale; UPDRS(I)—(IV), Movement Disorder Society—Unified Parkinson's Disease Rating Scale part 1 to 4; PDQ-39, Parkinson's Disease Questionnaire; MMSE, Mini-Mental State Examination; FAB, Frontal Assessment Battery; ESS, Epworth Sleepiness Scale; RBDQ, Rapid Eye Movement Sleep Behavior Disorder Questionnaire; Olfactory, Olfactory identification score; SCOPA-AUT, Scales for Outcomes in Parkinson's Disease-Autonomic; GDS-15, Geriatric Depression Scale; AS, Apathy Scale; NA, no data available.

\*median [lower–upper bound of interquartile range].

\*\*number of men: Women.

[a] Significant differences (p < 0.05) were found between the PD and control groups with the Mann–Whitney U test.

[b] Significant differences (p < 0.05) were found between the EDS+ and EDS- groups with the Mann–Whitney U test.

[c] Significant differences (p < 0.05) were found between the RBD+ and RBD- groups with the Mann–Whitney U test.

[d] Significant differences (p < 0.05) were found between the RTG and no DA groups with the Mann–Whitney U test.

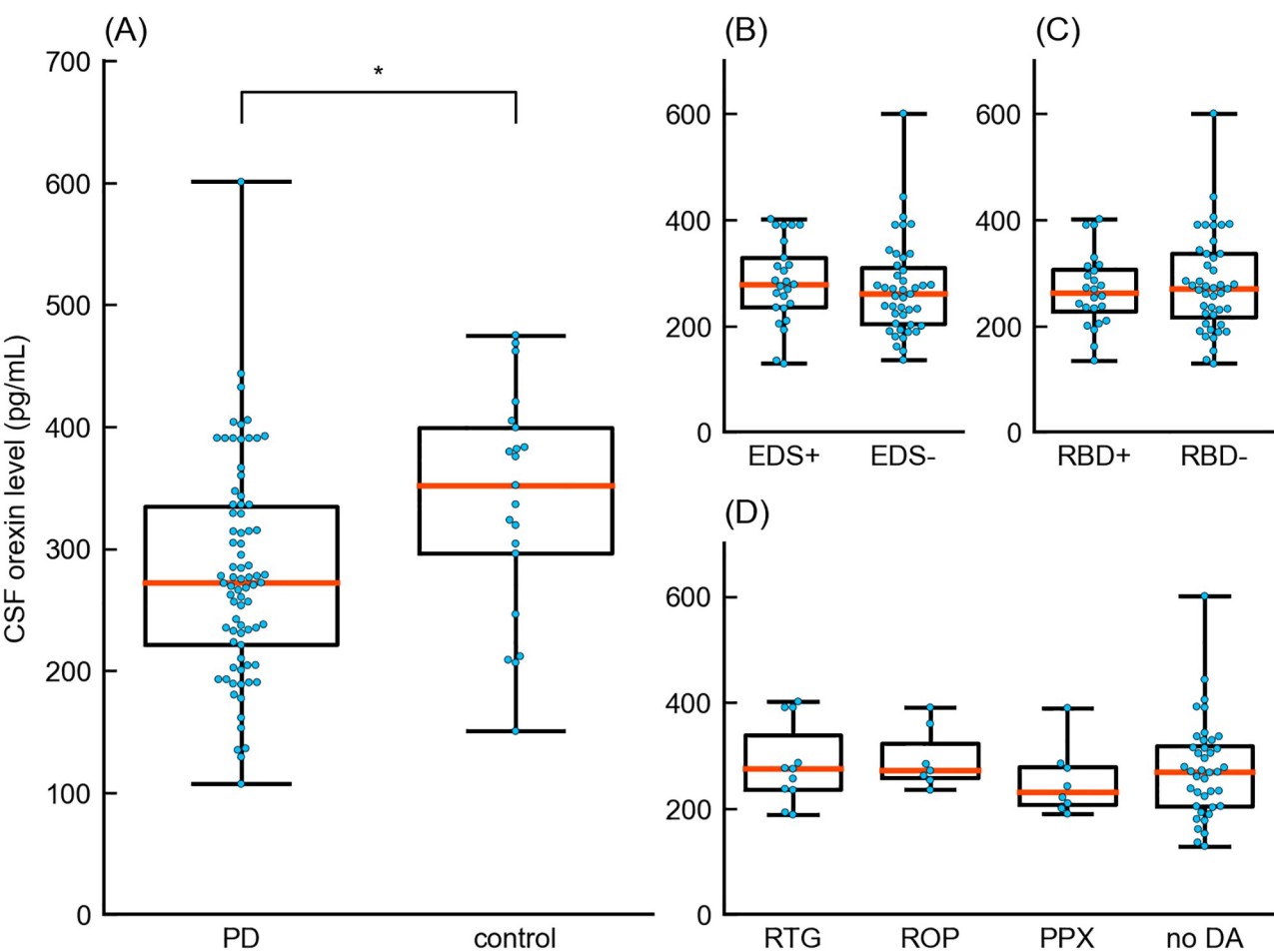

**Fig 2. Boxplot of CSF orexin levels.** Boxplots showing the comparison between (A) the PD and control group, (B) EDS+ and EDS- group, (C) RBD + and RBD- group, and (D) RTG, ROP, PPX and no DA groups. PD, Parkinson's disease; EDS, excessive daytime sleepiness; RBD, rapid eye movement sleep behavior disorder; RTG, rotigotine; ROP, ropinirole; PPX, pramipexole; no DA, no use of dopamine agonist; CSF, cerebrospinal fluid. * Significant differences were observed between the PD and control groups.

and PPX (231.7 [207.7–278.8] pg/mL) groups also showed no difference when compared with the noDA (269.8 [204.6–318.6] pg/mL) group (Fig 2 and Table 1).

The correlations between CSF orexin levels and sleep disturbances were also assessed. The correlation coefficients (ρ) between orexin and disease duration, ESS, and RBDQ were 0.198 (95% confidence interval (95%CI) = [-0.04–0.42], p = 0.11), 0.021 ([-0.22–0.26], p = 0.87) and 0.079 ([-0.16–0.31], p = 0.52), respectively (Fig 3). There were no statistically significant correlations between CSF orexin levels and other clinical characteristics (S1 Table).

Multiple linear regression showed no significant correlation between orexin levels and other clinical characteristics, including age, sex, disease duration, ESS, MDS-UPDRS part 3, and MMSE, with an adjusted $R^2$ of 0.078 (degrees of freedom = 6, F statistic = 1.94, p = 0.09). Regression coefficients (β) of clinical characteristics were 9.74 for age (95%CI = [-12.75–32.23], t statistic = 0.87, p = 0.39), -7.69 for sex ([-29.8–14.42], t = -0.7, p = 0.49), 22.36 for disease duration ([0.52–44.19], t = 2.05, p = 0.04), 1.02 for ESS ([-19.94–21.99], t = 0.1, p = 0.92), -21.49 for UPDRS (III) ([-43.63–0.64], t = -1.94, p = 0.06), and -15.56 for MMSE ([-40.39–9.26], t = -1.25, p = 0.22).

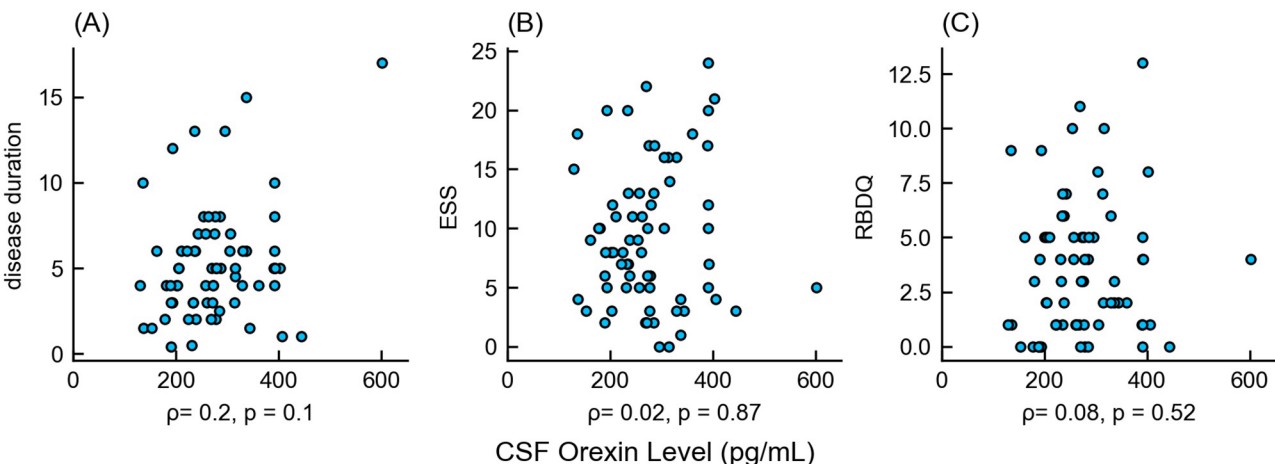

**Fig 3. Correlation between orexin levels and clinical characteristics in patients with Parkinson's disease.** In each scatter plot, the x-axis shows CSF orexin levels and the y-axis shows (A) disease duration, (B) ESS, and (C) RBDQ. The disease duration was scaled by year. All these clinical characteristics showed poor correlation with CSF orexin levels, and the correlation coefficients were not statistically significant. CSF, cerebrospinal fluid; ESS, Epworth Sleepiness Scale; RBDQ, Rapid Eye Movement Sleep Behavior Disorder Questionnaire.

### Orexin levels in clinical course of patients with Parkinson's disease

We analyzed 16 time-series sets of samples from 15 patients. One patient had two sets. Longitudinal changes in CSF orexin levels are shown in Fig 4. Among these 16 sets, 12 showed an increase in CSF orexin levels from initial examinations. The median of this increase in CSF orexin levels from baseline (38.1 pg/mL) was not statistically significant (95%CI = [-32.9–95.2], Z statistic = -1.84, p = 0.07, effect size r = -0.325).

There were no correlations between the longitudinal changes in CSF orexin levels and those of ESS ($\rho$ = 0.13, 95% CI = [-0.39–0.59], p = 0.63) or RBD ($\rho$ = 0.09 [-0.43–0.56], p = 0.74) (Fig 5); there were also no correlations between CSF orexin changes and other clinical characteristics. (S2 Table). No obvious relationship was found between initial CSF orexin levels and longitudinal changes in ESS ($\rho$ = 0.17 [-0.35–0.61], p = 0.53) and RBD ($\rho$ = -0.04 [-0.53–0.46], p = 0.88) (Fig 6).

## Discussion

To our knowledge, this is the study with the largest number of CSF samples of patients with PD. Here, we investigated the role of orexin in PD compared to controls, the relationship between orexin and various clinical characteristics, and the effect of orexin on longitudinal changes in PD. We revealed decreased CSF orexin levels in patients with PD, which were poorly related to any cross-sectional and longitudinal clinical characteristics in PD.

### Orexin in PD compared to controls

Our study showed that CSF orexin levels in patients with PD were lower than those in the controls. However, the pathological relevance of CSF orexin level needs to be considered. The normal range for CSF orexin level defined for the diagnosis of narcolepsy is as follows: over 200 pg/mL is normal, between 110 and 200 pg/mL in intermediate, and under 110 pg/mL is the abnormal [34]. In our study, the median CSF orexin level was within the normal range, and 64 out of 78 CSF samples (82.1%) showed a normal orexin range (over 200 pg/mL). Although one study showed that CSF orexin levels in patients with PD decreased to an intermediate range

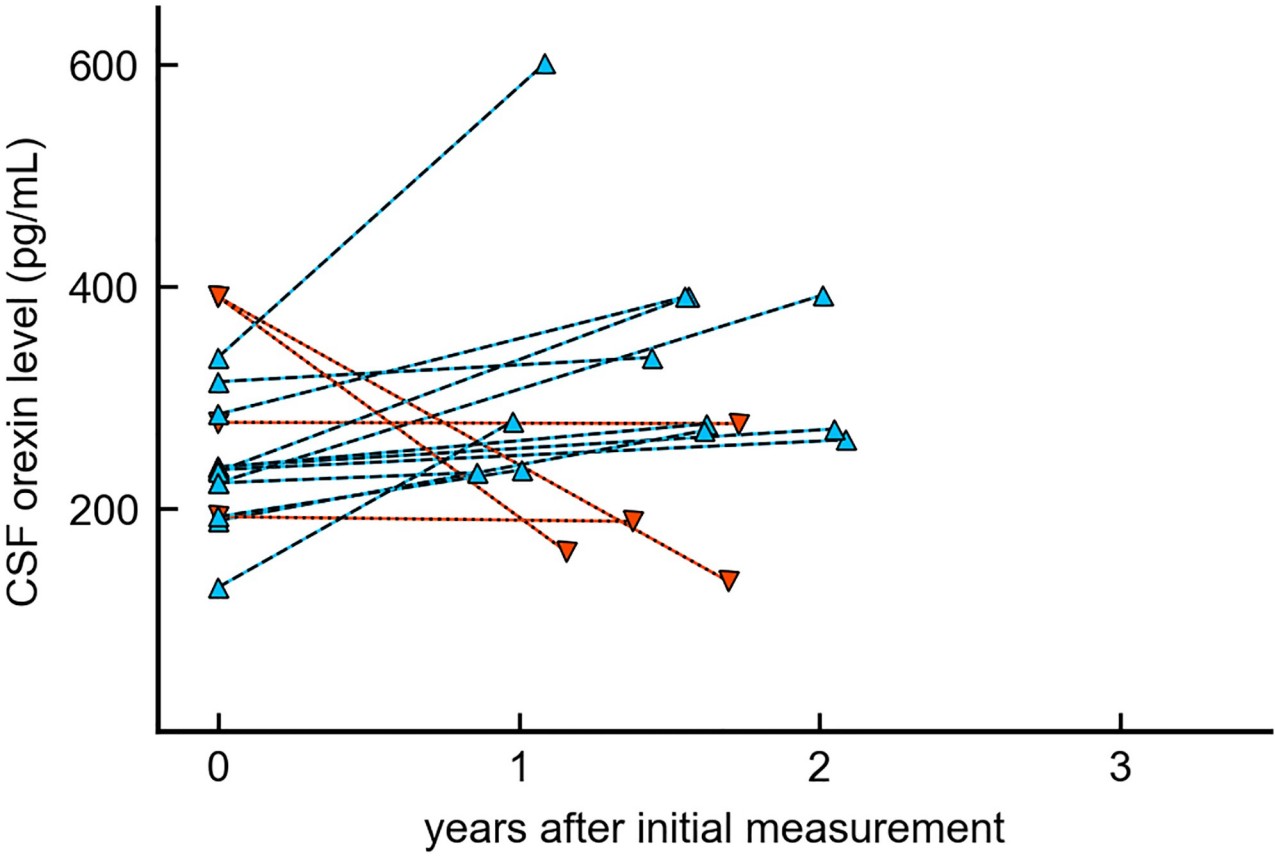

**Fig 4. Longitudinal changes in orexin levels in Parkinson's disease.** In this scatter plot, the x-axis shows the time period since the initial measurement and the y-axis shows CSF orexin levels. Blue triangles show 12 cases with increased CSF orexin levels after the interval, and red inverted triangles show 4 cases with decreased CSF orexin levels after the interval.

[13], other studies demonstrated normal CSF orexin levels in these patients [11,12]. In our study, CSF orexin levels in patients with PD were lower than those of controls, but this reduction was smaller than that in patients with narcolepsy.

## Correlation of orexin with clinical characteristics

First, our study showed that there was no significant difference in CSF orexin levels in the presence and absence of EDS. Second, no difference was noted in these levels between patients with and those without RBD. We did not observe a relationship between the severity of sleep disturbances and orexin levels using simple regression or multiple linear regression analysis. This study emphasizes the minute effect of orexin on EDS and its negative impact on RBD. The other clinical characteristics were not correlated. Previous studies on this correlation had conflicting outcomes; some reported no correlation between CSF orexin levels and ESS [12,14], but multiple sleep latency tests showed a positive correlation between CSF orexin levels and shortened sleep latency [21]. For the RBDQ, some studies showed a relationship with CSF orexin levels [19], but others did not [13].

In our study, the difference observed in CSF orexin levels between patients with PD and those with narcolepsy might be due to the pathological characteristics of these diseases. Pathological analysis showed that α-synuclein deposits that were seen in patients with PD were not

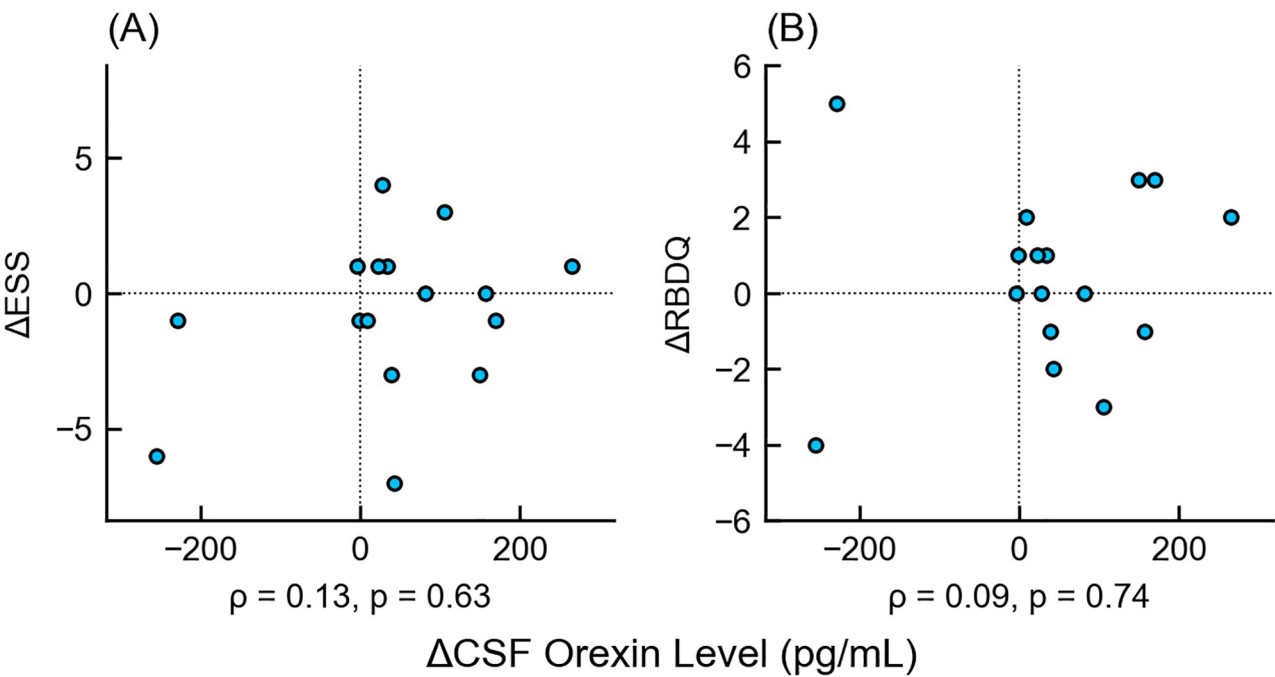

**Fig 5. Scatter plot of longitudinal changes in orexin levels and main clinical characteristics.** The x-axis shows the increase in CSF orexin levels since the initial examination, and the y-axis shows the increase in (A) ESS or (B) RBDQ since the initial examination. There was no correlation between the increase in CSF orexin levels and the clinical scores. Δ-, increment of -; ESS, Epworth Sleepiness Scale; RBDQ, Rapid Eye Movement Behavior Disorder Questionnaire.

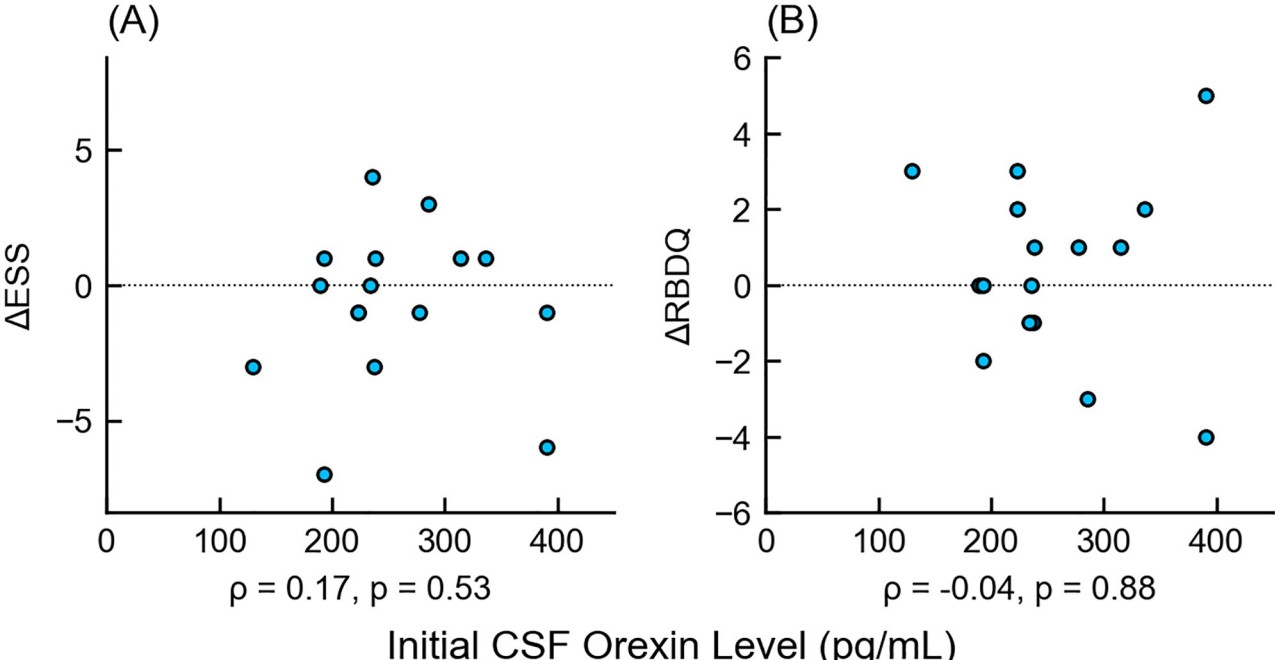

**Fig 6. Scatter plot of initial cerebrospinal fluid orexin levels and longitudinal changes in main clinical characteristics.** The x-axis shows the initial CSF orexin levels, and the y-axis shows the increase in (A) ESS or (B) RBDQ from the initial examination. There was no correlation between the initial CSF orexin levels and the increase in clinical scores. Δ-, increment of -; ESS, Epworth Sleepiness Scale; RBDQ, Rapid Eye Movement Behavior Disorder Questionnaire.

found in those with narcolepsy, which might have caused a different trend in video-polysomnography analysis for RBD in PD and narcolepsy [35]. Neuronal loss in narcolepsy is limited to orexin neurons, but PD is a neurodegenerative disease, where broad neuronal systems are impaired, including not only orexin neurons, but also noradrenaline, serotonin, and acetylcholine neurons. These neurotransmitters also play important roles in sleep-arousal mechanisms [36]. Furthermore, orexin fibers project widely to the nuclei in the brainstem and hypothalamus, including the locus coeruleus and raphe nuclei [37]. These nuclei are damaged in PD [38], which may cause dysfunction of the orexin system without damaging orexin neurons through a decrease in orexin receptors. Therefore, CSF orexin levels are poorly correlated to clinical sleep disturbances in PD.

Our study showed no significant differences in CSF orexin levels between patients with PD with and those without DAs. Since our study was retrospective, it is possible that DA had already been discontinued owing to adverse effects. A previous study showed an improvement in EDS and an increase in CSF orexin levels after switching from PPX to pergolide [14]. In clinical situations, the effect of DA use on orexin levels seems to be small, but further studies are needed to focus on cases with adverse sleep events.

Taking all the results together, we believe that the decreased orexin level in patients with PD does not exert a substantial effect on the PD pathologies, at least at the relatively early stage that we have examined.

### Effect of orexin on longitudinal changes in PD

Our study showed no significant correlation between CSF orexin levels and PD disease duration, which is in line with a previous study [15]; however, another study demonstrated a negative correlation between CSF orexin levels and disease duration [14]. This discrepancy in results could be due to differences in study conditions such as patient numbers.

This study focused on the relationship between longitudinal changes in CSF orexin levels and clinical characteristics. Our study revealed that the CSF orexin level was a poor predictor of the clinical course, including sleep disturbances in PD. A previous study in rats revealed that 73% of orexin neurons are lost when CSF orexin levels from the cisterna magna are reduced to half, and CSF orexin levels are not related to the decline in orexin neurons [39]. According to a previous report, patients with advanced PD (HYscale score = 4) still have more than half the number of orexin cells compared to those in healthy controls [7]; therefore, the decrease in orexin neurons may not appear in the CSF until the advanced stage of PD. In our study, the intervals from the initial examinations were no more than three years, which may be too short to observe the progression of PD and changes in the orexin system.

### Limitations

Our study had several limitations. First, it was retrospective in nature, which might have caused selection bias. Second, EDS and RBD were evaluated only by a subjective questionnaire, so they may not reflect the objective sleep period and rhythms. Third, we included only a few patients with advanced PD, and the intervals from the initial examinations were within three years. The absence of long-term data has led to biased analysis. Fourth, we recruited patients who were not diagnosed with neurodegenerative diseases, including those with normal pressure hydrocephalus, as controls. These patients potentially differ from the healthy elderly; however, patients with these diseases were assigned to control groups in a previous study [40] and were shown to have normal orexin levels [41]. Fifth, all patients had optimal medication during examinations; therefore, we could not rule out the impact of dopamine replacement

therapy. Since DAs, including levodopa, affect circadian regulation, we could not eliminate such effects from the drugs [2].

## Conclusion

Although orexin is important for normal sleep regulation, our results emphasized that the efficacy of CSF orexin levels as a biomarker for PD is limited. Since many nervous systems are impaired in PD, the relationship with other neurotransmitters should also be evaluated. To determine the impact of orexin levels on PD, it may be necessary to further evaluate drug-naive patients or advanced patients with significant orexin neuronal loss. Understanding the reciprocal role of orexin among other neurotransmitters may provide a better treatment strategy for sleep disturbance, which may contribute to improving the quality of life of patients with PD.

## Supporting information

**S1 Table. Correlation between cerebrospinal fluid orexin levels and clinical characteristics in Parkinson's disease.** The table shows Spearman's rank correlation coefficients for orexin levels and clinical characteristics. No clinical characteristics showed a significant correlation with CSF orexin levels. 95%CI, 95% confidence interval; ESS, Epworth Sleepiness Scale; RBDQ, Rapid Eye Movement Sleep Behavior Disorder Questionnaire; HYscale, Hoehn and Yahr scale; MDS-UPDRS(I)—(IV), Movement Disorder Society—Unified Parkinson's Disease Rating Scale part 1 to 4; PDQ-39, Parkinson's Disease Questionnaire; MMSE, Mini-Mental State Examination; FAB, Frontal Assessment Battery; Olfactory, Olfactory identification score; SCOPA-AUT, Scales for Outcomes in Parkinson's Disease-Autonomic; GDS-15, Geriatric Depression Scale; AS, Apathy Scale.
(XLSX)

**S2 Table. Correlation between longitudinal changes orexin levels and clinical characteristics in Parkinson's disease.** The table shows the Spearman's rank correlation coefficients for longitudinal changes in orexin levels and clinical characteristics. There were no significant correlations between longitudinal changes in orexin levels and clinical characteristics. 95%CI, 95% confidence interval; Δ-, increment of -; ESS, Epworth Sleepiness Scale; RBDQ, Rapid Eye Movement Sleep Behavior Disorder Questionnaire; HYscale, Hoehn and Yahr scale; MDS-UPDRS(I)—(IV), Movement Disorder Society—unified Parkinson's Disease Rating Scale part 1 to 4; PDQ-39, Parkinson's Disease Questionnaire; MMSE, Mini-Mental State Examination; FAB, Frontal Assessment Battery; Olfactory, Olfactory identification score; SCOPA-AUT, Scales for Outcomes in Parkinson's Disease-Autonomic; GDS-15, Geriatric Depression Scale; AS, Apathy Scale.
(XLSX)

**S3 Table. Correlation between initial cerebrospinal fluid orexin levels and longitudinal changes in clinical characteristics in Parkinson's disease.** The table shows Spearman's rank correlation coefficients for initial orexin levels and longitudinal changes in clinical characteristics. There was no significant correlation between initial CSF orexin levels and longitudinal changes in clinical characteristics. 95%CI, 95% confidence interval; Δ-, increment of -; ESS, Epworth Sleepiness Scale; RBDQ, Rapid Eye Movement Sleep Behavior Disorder Questionnaire; HYscale, Hoehn and Yahr scale; MDS-UPDRS(I)—(IV), Movement Disorder Society—unified Parkinson's Disease Rating Scale part 1 to 4; PDQ-39, Parkinson's Disease Questionnaire; MMSE, Mini-Mental State Examination; FAB, Frontal Assessment Battery; Olfactory, Olfactory identification score; SCOPA-AUT, Scales for Outcomes in Parkinson's Disease-

Autonomic; GDS-15, Geriatric Depression Scale; AS, Apathy Scale.
(XLSX)

## Author Contributions

**Conceptualization:** Takuya Ogawa, Yuta Kajiyama, Takashi Kanbayashi, Kensuke Ikenaka, Hideki Mochizuki.

**Data curation:** Takuya Ogawa, Gajanan S. Revankar, Tomohito Nakano.

**Formal analysis:** Takuya Ogawa, Gajanan S. Revankar, Tomohito Nakano, Seira Taniguchi.

**Funding acquisition:** Hideki Mochizuki.

**Investigation:** Takuya Ogawa, Yuta Kajiyama, Kensuke Ikenaka.

**Methodology:** Takuya Ogawa, Yuta Kajiyama, Gajanan S. Revankar, Tomohito Nakano, Kensuke Ikenaka.

**Project administration:** Yuta Kajiyama, Kensuke Ikenaka.

**Resources:** Hideaki Ishido, Shigeru Chiba, Takashi Kanbayashi, Kensuke Ikenaka.

**Supervision:** Takashi Kanbayashi, Kensuke Ikenaka, Hideki Mochizuki.

**Validation:** Takuya Ogawa.

**Visualization:** Takuya Ogawa.

**Writing – original draft:** Takuya Ogawa.

**Writing – review & editing:** Yuta Kajiyama, Gajanan S. Revankar, Seira Taniguchi, Kensuke Ikenaka, Hideki Mochizuki.

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
