## [Decision Letter · Decision Letter 0]

21 Nov 2022

PONE-D-22-21585Decreased  cerebrospinal fluid orexin levels not associated with clinical sleep disturbance in Parkinson’s disease: A retrospective studyPLOS ONE

Dear Dr. Ogawa,

Thank you for submitting your manuscript to PLOS ONE. After careful consideration, we feel that it has merit but does not fully meet PLOS ONE’s publication criteria as it currently stands. Therefore, we invite you to submit a revised version of the manuscript that addresses the points raised during the review process.

 Please review the manusript giving more informations about novelty and clinical significance of the results proposed.

We look forward to receiving your revised manuscript.

Kind regards,

Claudio Liguori

Academic Editor

PLOS ONE

Journal Requirements:

Additional Editor Comments (if provided):

The Authors proposed an interesting study about CSF orexin levels in patients with PD. Several studies already proposed the same topic and Authors should better delineate the novelty of their research. The main criticism is the lack of pathological reduction of CSF orexin levels in patients than controls, although CSF orexin levels at the mere statistical analysis resulted lower. Authors should better discuss their results.

Reviewers' comments:

Reviewer's Responses to Questions

**Comments to the Author**

1. Is the manuscript technically sound, and do the data support the conclusions?

Reviewer #1: Yes

2. Has the statistical analysis been performed appropriately and rigorously? 

Reviewer #1: Yes

3. Have the authors made all data underlying the findings in their manuscript fully available?

Reviewer #1: Yes

4. Is the manuscript presented in an intelligible fashion and written in standard English?

Reviewer #1: Yes

5. Review Comments to the Author

Reviewer #1: Understanding the role of orexin in Parkinson’s disease (PD) is an intriguing challenge, with both physiopathological and therapeutic implications.

In this retrospective study Authors tried to investigate a possible alteration of orexin levels in PD. Authors also looked for possible correlations between altered orexin levels and sleep disorders / clinical characteristics of the pathology.

I appreciated Authors’ purpose and study design. The introduction of the study rightly takes into account several controversial findings.

Clear rationale, appropriate methods and analysis, well-argued introduction and discussion.

The major finding of this study, according to several previous reports, is that PD patients showed decreased CSF orexin levels. However, orexin levels showed no correlation with any clinical characteristic, that is, neither with sleep disorders nor with disease gravity/progression.

I greatly appreciate the large number of CSF samples of PD patients, that in my opinion is a great strength of this study.

The same authors well explained the limitations of the study, which do not interfere with the scientific validity of the study.

Moreover, it is a well written article.

6. PLOS authors have the option to publish the peer review history of their article (what does this mean?). If published, this will include your full peer review and any attached files.

Reviewer #1: **Yes: **Rocco Cerroni

---

## [Author Response · Author response to Decision Letter 0]

12 Dec 2022

Response to Reviewer and Editor

#1 Journal Requirements

We note that you have stated that you will provide repository information for your data at acceptance. Should your manuscript be accepted for publication, we will hold it until you provide the relevant accession numbers or DOIs necessary to access your data. If you wish to make changes to your Data Availability statement, please describe these changes in your cover letter and we will update your Data Availability statement to reflect the information you provide.

Answer to #1

Thank you for the notion. Our data is now available in the Dryad repository (https://datadryad.org/stash/share/IFCMVjwCZFzhhIQncweMKlKNH0pP4Fs1h3khfr0VAD0), as we described in the online submission system. The dataset will be available in the citation below after publication.

(Ogawa, Takuya et al. (2022), dataset_of_CSF_orexin_level_in_PD_with_clinical_assessment, Dryad, Dataset, https://doi.org/10.5061/dryad.905qfttq0)

#2 Additional Editor Comments

The Authors proposed an interesting study about CSF orexin levels in patients with PD. Several studies already proposed the same topic and Authors should better delineate the novelty of their research. The main criticism is the lack of pathological reduction of CSF orexin levels in patients than controls, although CSF orexin levels at the mere statistical analysis resulted lower. Authors should better discuss their results.

Answer to #2

We agree with your point regarding the lack of pathological reduction of CSF orexin levels in patients compared to controls, although CSF orexin levels were statistically lower. We already mentioned the novelty in terms of the large sample and various sufficient clinical evaluations. We also added a new sentence to the introduction section (p. 6 Lines 105-107) to clarify our motivation for this study. We also reconstructed the discussion section to describe the minimal pathological meaning of a decreased CSF orexin level in PD compared to controls (p. 18, Lines 350-351; p. 18, Lines 355-357; p. 20 Lines 398-400).

(Section 1. Introduction) Page 6, lines 105-107:

“Since previous studies had insufficient samples or used inconsistent methods of clinical assessment [10,11,13,14,16,18,19,21], the diagnostic or prognostic value of the CSF orexin level in PD lacks evidence. In this study, we measured orexin levels in the CSF samples obtained from patients with PD to clarify the role of orexin as a biomarker of PD, in the following particulars:...”

(Section 2. Discussion > Orexin in PD compared to controls) Page 18, lines 348

“Our study showed that CSF orexin levels in patients with PD were lower than those in the controls. However, the pathological relevance of CSF orexin levels needs to be considered. The normal range for CSF orexin level defined for the diagnosis of narcolepsy is as follows: over 200 pg/mL is normal, between 110 and 200 pg/mL is intermediate, and under 110 pg/mL is abnormal [34].”

Page 18, lines 354-356

“Although one study showed that CSF orexin levels in patients with PD decreased to an intermediate range [13], other studies demonstrated normal CSF orexin levels in these patients [11,12].”

(Section 3. Discussion > Correlation of orexin with clinical characteristics) Page 20, lines 395-397

“Our study showed no significant differences in CSF orexin levels between patients with PD with and those without DAs. Since our study was retrospective, it is possible that DA had already been discontinued owing to adverse effects. A previous study showed an improvement in EDS and an increase in CSF orexin levels after switching from PPX to pergolide [14]. In clinical situations, the effect of DA use on orexin levels seems to be small, but further studies are needed to focus on cases with adverse sleep events.

Taking all the results together, we believe that the decreased orexin level in patients with PD does not exert a substantial effect on the PD pathologies, at least at the relatively early stage that we have examined.“

---

## [Editor Report · Decision Letter 1]

15 Dec 2022

Decreased  cerebrospinal fluid orexin levels not associated with clinical sleep disturbance in Parkinson’s disease: A retrospective study

PONE-D-22-21585R1

Dear Dr. Ogawa,

We’re pleased to inform you that your manuscript has been judged scientifically suitable for publication and will be formally accepted for publication once it meets all outstanding technical requirements.

Kind regards,

Claudio Liguori

Academic Editor

PLOS ONE
---

## [Editor Report · Acceptance letter]

21 Dec 2022

PONE-D-22-21585R1 

Decreased cerebrospinal fluid orexin levels not associated with clinical sleep disturbance in Parkinson’s disease: A retrospective study 

Dear Dr. Ikenaka:

I'm pleased to inform you that your manuscript has been deemed suitable for publication in PLOS ONE. Congratulations! Your manuscript is now with our production department. 

Kind regards, 

on behalf of

Dr. Claudio Liguori 

Academic Editor

PLOS ONE